# Visible-to-mid-IR tunable frequency comb in nanophotonics

Arkadev Roy[1,3], Luis Ledezma [1,2,3], Luis Costa [1], Robert Gray[1], Ryoto Sekine [1], Qiushi Guo[1], Mingchen Liu[1], Ryan M. Briggs [2] & Alireza Marandi [1]✉

Optical frequency comb is an enabling technology for a multitude of applications from metrology to ranging and communications. The tremendous progress in sources of optical frequency combs has mostly been centered around the near-infrared spectral region, while many applications demand sources in the visible and mid-infrared, which have so far been challenging to achieve, especially in nanophotonics. Here, we report widely tunable frequency comb generation using optical parametric oscillators in lithium niobate nanophotonics. We demonstrate sub-picosecond frequency combs tunable beyond an octave extending from 1.5 up to 3.3 μm with femtojoule-level thresholds on a single chip. We utilize the up-conversion of the infrared combs to generate visible frequency combs reaching 620 nm on the same chip. The ultra-broadband tunability and visible-to-mid-infrared spectral coverage of our source highlight a practical and universal path for the realization of efficient frequency comb sources in nanophotonics, overcoming their spectral sparsity.

Optical frequency combs consisting of several spectral lines with accurate frequencies are at the core of a plethora of modern-day applications[1,2], including spectroscopy[3], optical communication[4], optical computing[5], atomic clocks[6], ranging[7,8] and imaging[9]. Many of these applications demand optical frequency combs in the technologically important mid-infrared[10,11] and visible[12,13] spectral regimes. Accessing optical frequency comb sources in integrated photonic platforms is of paramount importance for the translation of many of these technologies to real-world applications and devices[14]. Despite outstanding progress in that direction in the near-infrared, there is a dearth of widely tunable frequency comb sources, especially in the highly desired mid-infrared and visible spectral regimes.

Notable efforts on miniaturized mid-IR comb sources typically rely on supercontinuum generation and/or intra-pulse difference frequency generation[15,16]. Not only do these nonlinear processes usually require a femtosecond pump as an input (which has its own challenges for efficient on-chip manifestation), but their power is also distributed over a wide frequency range, including undesired spectral bands. While the broad instantaneous spectral bandwidth may be suitable for

certain applications, others may require the existence of more concentrated spectral power to enhance the signal-to-noise ratio. Engineered semiconductor devices like quantum cascade lasers have successfully been demonstrated as mid-infrared frequency comb sources[17], however, they are not tunable over a broad wavelength range and are still difficult to operate in the ultrashort pulse regime[18,19]. The situation is exacerbated by the lack of a suitable laser gain medium that is amenable to room temperature operation in the mid-IR. Kerr nonlinearity can lead to tunable broadband radiation[20–22] but is contingent on satisfying demanding resonator quality factor requirements and typically relies on a mid-IR pump, to begin with, for subsequent mid-infrared frequency comb generation. Similar challenges exist for Raman-based mid-IR frequency comb generation[23].

On the other hand, optical parametric oscillators (OPOs) based on quadratic nonlinearity have been the predominant way of accessing tunable coherent radiation in the mid-IR spectral region enjoying broadband tunability through appropriate phase matching of the three-wave mixing[11]. However, their impressive generation of tunable frequency combs in the mid-infrared has been limited to bulky free-

[1]Department of Electrical Engineering, California Institute of Technology, Pasadena, California 91125, USA. [2]Jet Propulsion Laboratory, California Institute of Technology, Pasadena, California 91109, USA. [3]These authors contributed equally: Arkadev Roy, Luis Ledezma. ✉e-mail: marandi@caltech.edu

space configurations pumped by femtosecond lasers[24,25]. Recently, integrated quadratic OPOs have been realized in the near-IR, using high-Q resonators with pump-resonant designs[26–28], which have not been able to access the broad tunability of phase matching and mid-IR frequency comb generation.

In this work, we demonstrate ultra-widely tunable frequency comb generation from on-chip OPOs in lithium niobate nanophotonics. Leveraging the ability to control the phase-matching via periodic poling combined with dispersion engineering, we show an on-chip tuning range that exceeds an octave. We pump the OPOs with picosecond pulses from an electro-optic frequency comb source in the near-IR, which is already demonstrated to be compatible with nanophotonic lithium niobate[29–31]. The demonstrated frequency combs cover the typical communication bands and extend into the mid-infrared spectral region beyond 3 μm with instantaneous bandwidths supporting sub-picosecond pulse durations. Additionally, the same chip produces tunable frequency combs in the visible resulting from up-conversion processes. Tunable visible frequency comb realization has been challenging owing to the absence of a suitable broadband gain medium and the typical large normal dispersion at these wavelengths in most integrated photonic platforms[32,33]. Our demonstration from a single chip also gains importance from the fact that the approach is amenable to wafer-scale fabrication with uniform process steps, thereby paving the way for mass-producible integrated tunable frequency comb sources.

## Results

To achieve broadband and widely tunable frequency combs, we design a doubly resonant OPO[34–36] based on nano-waveguides etched on X-cut 700-nm-thick MgO-doped lithium niobate, which is illustrated in Fig. 1a. Unlike the previous triply-resonant designs[26–28], our design provides access to the wide tunability of quasi-phase-matching (QPM) and avoids stringent requirements such as ensuring the resonance of the pump[37]. Doubly resonant operation is achieved by controlling the precise spectral response of the OPO resonator using two spectrally selective adiabatic couplers (highlighted in Fig. 1a) that only let the long wavelengths (signal and idler) to resonate in the OPO while allowing the short wavelengths (pump and up-converted light) to leave the cavity (see Supplementary Section 3). This is not only important for achieving a broad tuning range for the signal and the idler, but it also enables non-resonant broadband and widely tunable up-conversion into the visible, which is in stark contrast with previous parametric sources in that range[32,33]. Another important aspect of the on-chip OPO design is the dispersion engineering of the main interaction waveguide of the OPO, which in combination with periodic poling, leads to broad spectral coverage of the QPM tuning. Engineering the dispersion of the remainder of the resonator is another important design degree of freedom that can be further utilized for achieving quadratic solitons and pulse compression mechanisms[38].

Quadratic parametric nonlinear interactions take place in a 5-mm-long poled waveguide region, which has a fixed poling period (Λ) for each OPO on the chip. The periodic poling phase-matches parametric nonlinear interaction between the pump, the signal, and the idler waves, which can be tuned from degeneracy to far non-degeneracy. The chip consists of multiple OPOs with poling periods for type-0 phase matching of down-conversion of a non-resonant pump at around 1 μm to an octave-spanning range of resonant signal and idler wavelengths, i.e., the OPO output. The QPM tuning curves are shown in Fig. 1b. In addition to these OPO outputs, the poled waveguide also provides additional parametric up-conversion processes, notably the second-

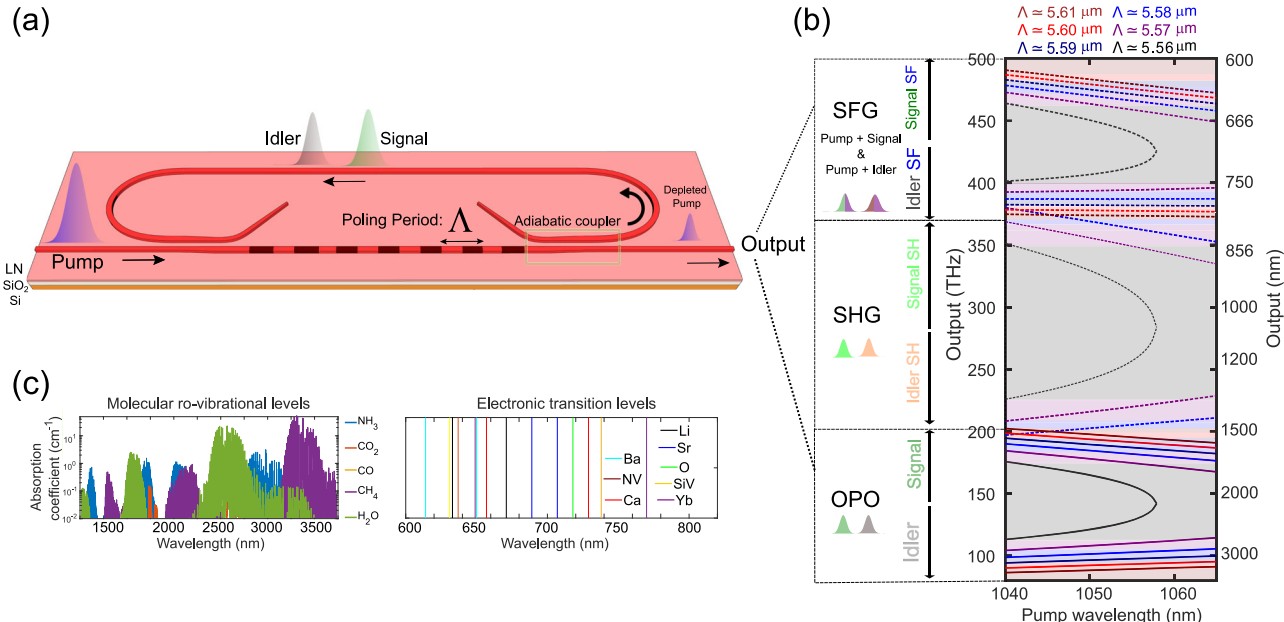

**Fig. 1 | Ultra-widely tunable frequency combs from nanophotonic parametric oscillators. a** Schematic of a doubly resonant optical parametric oscillator fabricated on an X-cut thin-film lithium niobate consisting of a periodically poled region for efficient parametric nonlinear interaction. The waveguides (dimensions: width of 2.5 μm, etch depth of 250 nm) support guided-modes in the mid-infrared corresponding to the idler wave. **b** Quasi-phase matched parametric gain tuning from visible-to-mid-IR. Phase-matching curves leading to tunable mid-infrared idler emission enabled by optical parametric oscillator devices with slightly different poling periods (Λ) integrated on the same chip. The same chip is capable of producing tunable visible frequency combs thanks to the sum-frequency generation (SFG) process between the pump with the signal and idler waves. Other

accompanying up-conversion processes include the second-harmonic (SH) of the signal and the idler. The phase-matching curves for the up-conversion processes are plotted (dotted lines) according to the energy conservation relations and do not strictly satisfy quasi-phase-matching. Some second-harmonic phase-matching curves have been omitted for better clarity. **c** The emission from the chip overlaps with strong molecular absorption lines in the mid-infrared, covering a spectral window important for molecular spectroscopy. The spectral coverage in the visible includes atomic transition wavelengths corresponding to commonly used trapped ions/ neutral atoms/color centers.

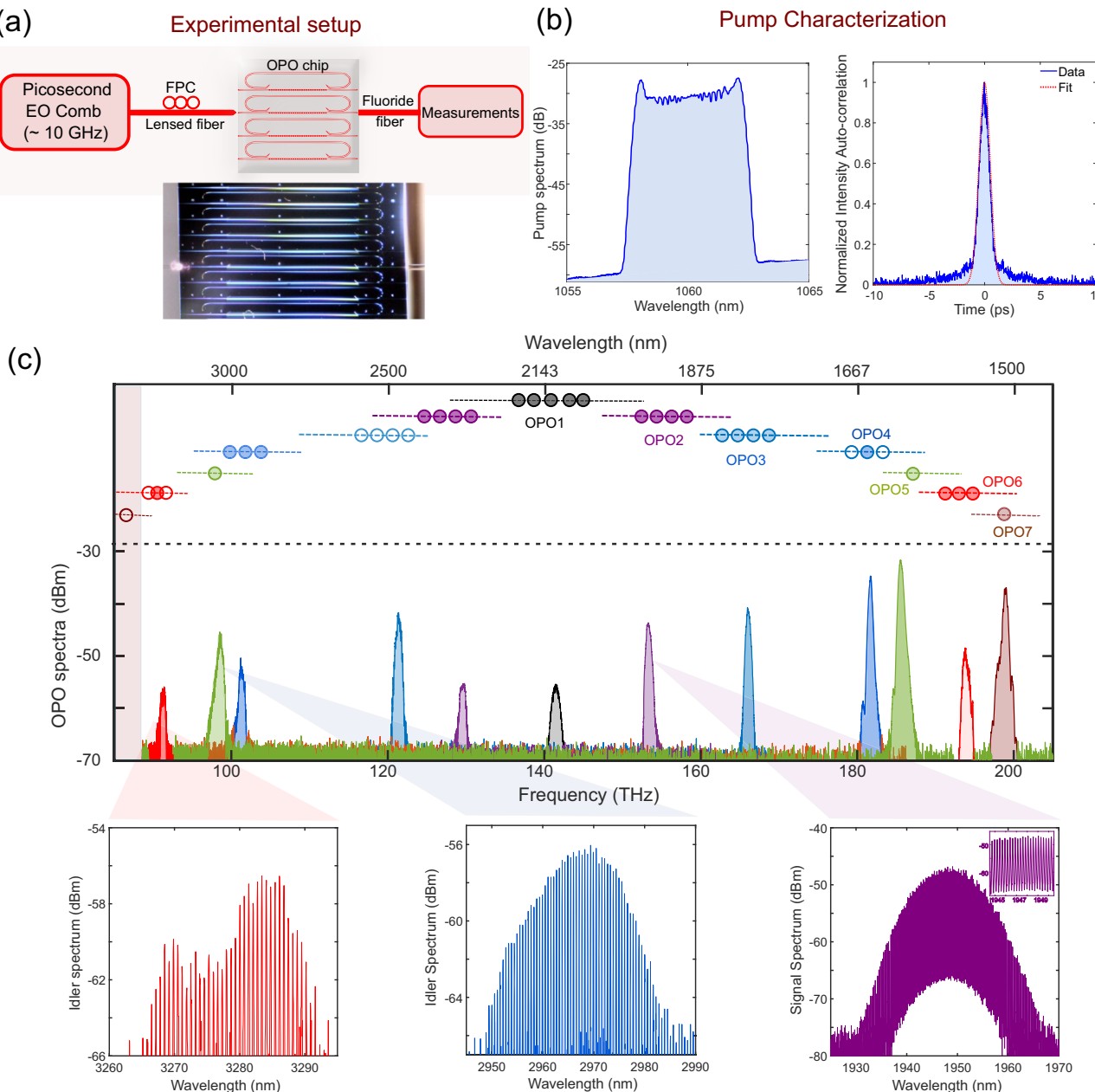

**Fig. 2 | Near-IR to mid-IR frequency combs from nanophotonic OPOs on a single chip. a** Schematic of the experimental setup used to pump and measure the synchronously pumped optical parametric oscillator chip. The image of the OPO chip is shown alongside, **b** Experimental measurements of the spectral and temporal characteristics (intensity auto-correlation trace) of the electro-optic pulsed pump showing a pulse-width of ~1 ps, **c** Broadband infrared spectral coverage of the OPO chip showing the signal and the idler spectrum as its operation is tuned from degeneracy to far non-degeneracy. Separate colors represent outputs from different OPO devices on the same chip with distinct poling periods. Zoomed-in versions display the underlying comb line structure of the power spectrum envelope of the signal and idler combs. The top panel represents the fine-tuning range of the corresponding OPO spectrum obtained by pump wavelength tuning. The dashed line indicates the tuning range for each OPO (assuming 30 nm of pump tuning). Filled dots represent measured data, while empty dots indicate inferred spectrum from its signal/idler counterpart. (Detailed spectrum for the tuning results are presented in Fig. 3c and Supplementary Section 14).

harmonic of the signal/idler and the sum-frequency generation from the pump and signal/idler. The overall tuning range of the chip overlaps with many molecular and atomic transitions, as illustrated in Fig. 1c. The strong spatio-temporal confinement of the interacting waves in the waveguide guarantees substantial up-conversion efficiencies, which can be further enhanced with the addition of proper poling periods and tailoring to specific applications (see Supplementary Section 2).

As shown in Fig. 1b, to continuously cover the visible to the mid-infrared, we focus on tuning the QPM by coarsely switching the poling period as well as fine-tuning the pump wavelength over ~25 nm. It is

worth noting that this tuning range for the pump is compatible with the existing semiconductor lasers[39]. Moreover, the coarse switching of the poling period can be achieved without mechanical movements, for instance, by means of electro-optic routing (see Supplementary Section 12). In addition, temperature tuning of the poled region can provide another substantial tuning mechanism (see Supplementary Section 13).

The OPO is synchronously pumped[35,40,41] by ~1-ps-long pulses operating at a repetition rate of approximately 19 GHz. The repetition rate was tuned close to the OPO cavity free spectral range or its

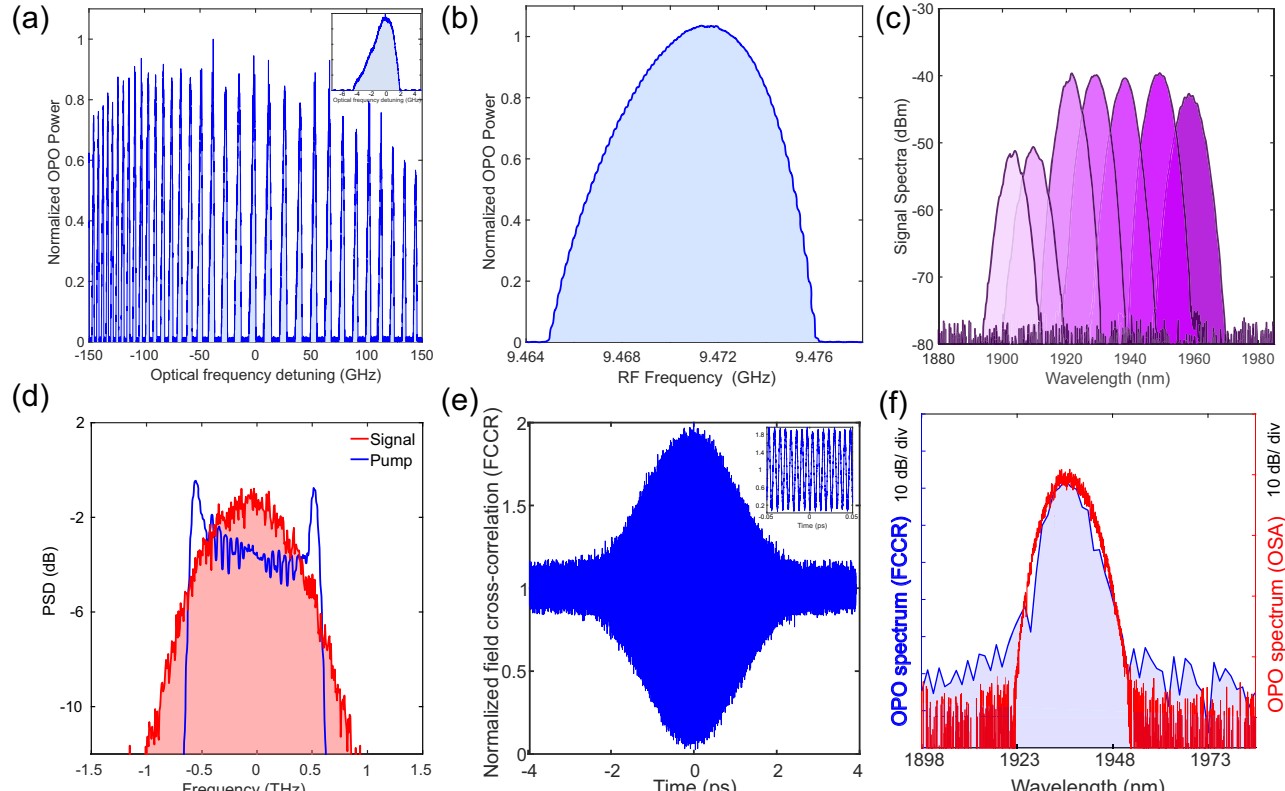

**Fig. 3 | Characteristics of the frequency comb generated from the synchro-nously pumped on-chip OPOs. a** Resonance peak structure obtained by sweeping the pump central wavelength which is typical of doubly resonant OPO operation. A zoomed-in view of a single peak is shown in the inset, **b** Range of the existence of the synchronously pumped OPO for a fixed pump power as the pump repetition rate is varied, **c** An example of fine-tuning of a single OPO output by tuning the pump central wavelength (please see Supplementary Section 14 for more experi-mental results), **d** Spectral broadening of the OPO operating at degeneracy corresponding to a sub-picosecond transform-limited duration of ~400 fs. The spectrum for both the pump and the signal are normalized for the aid of visuali-zation of the spectral broadening, **e** Verification of the coherence of the OPO output as evident from the existence of interference fringes (see inset) in the electric-field cross-correlation trace, **f** The close agreement between the spectra obtained from an optical spectrum analyzer measurement and that obtained by Fourier transforming the field cross-correlation corroborates the coherence of the OPO output.

harmonics (see Supplementary Section 6). The octave-wide tunability of the parametric oscillation from the OPO chip is obtained by tuning the pump central wavelength between 1040 and 1065 nm only. The pump is generated from an electro-optic frequency comb[42] (see Sup-plementary Section 4). The schematic of the experimental setup is shown in Fig. 2a. The spectral and temporal characteristics of the near-infrared pump are shown in Fig. 2b.

Figure 2c shows the broad spectral coverage of the OPO output extending up to 3.3 μm in the mid-infrared obtained from a single chip. The comb lines can be resolved by the optical spectrum analyzer (OSA) and can be seen in the inset, where the separation of the peaks cor-responds to the pump repetition rate. The on-chip threshold amounts to approximately 1 mW of average power (~50 mW of peak power and ~100 femtojoules of pulse energy) for the near-degenerate OPOs. The signal conversion efficiency approaches ~5% for the near-degenerate OPOs, while the mid-infrared (3.3 μm) idler conversion efficiency exceeds 1% for the far non-degenerate OPOs (see Supplementary Section 3). This corresponds to an estimated ~25 mW of peak power and ~5 μW of power per comb line in the mid-infrared (see Supple-mentary Section 3).

The doubly resonant operation of the OPO is also confirmed by the appearance of the resonance peak structure with the variation of the pump central wavelength, as shown in Fig. 3a. Figure 3b shows the tolerance of the synchronous pumping repetition rate mismatch with respect to the optimum OPO operating point. The fine tunability of the OPO output spectra, as offered by tuning the pump wavelength, is depicted in Fig. 3c. The combination of fine tunability and course tunability potentially enables continuous spectral coverage across the accessible spectral region. The OPO output at degeneracy (Fig. 3d) corresponds to a sub-picosecond transform-limited temporal duration (~400 fs), representing a pulse compression factor exceeding 2 with respect to the pump (see Supplementary Section 9). We further eval-uate the coherence of the output frequency comb by performing a linear field cross-correlation of the output signal light, as shown in Fig. 3e, where each OPO pulse is interfered with another pulse delayed by 10 roundtrips. The presence of the interference fringes (see inset of Fig. 3e), combined with the consistency of the Fourier transform of the cross-correlation trace and the signal spectrum obtained using an OSA, serve as evidence for the coherence of the output frequency comb over the entire spectrum (see Fig. 3f).

The occurrence of other quadratic nonlinear processes, namely second-harmonic generation (SHG) and sum-frequency generation (SFG), leads to frequency comb formation in the visible spectral region. The complete emission spectrum of an OPO consisting of the second harmonic of the pump and the signal waves, the sum-frequency components between the pump and the signal/idler waves, along with the usual signal/idler is shown in Fig. 4a. The scat-tered visible light emanating from the chip is captured by the optical microscope image (see Fig. 4b) showing the emission of pump second harmonic (green) and the sum-frequency components (red). Note that in the poling region, green dominates at the input side, which pro-gressively is overpowered by the sum-frequency red component. The

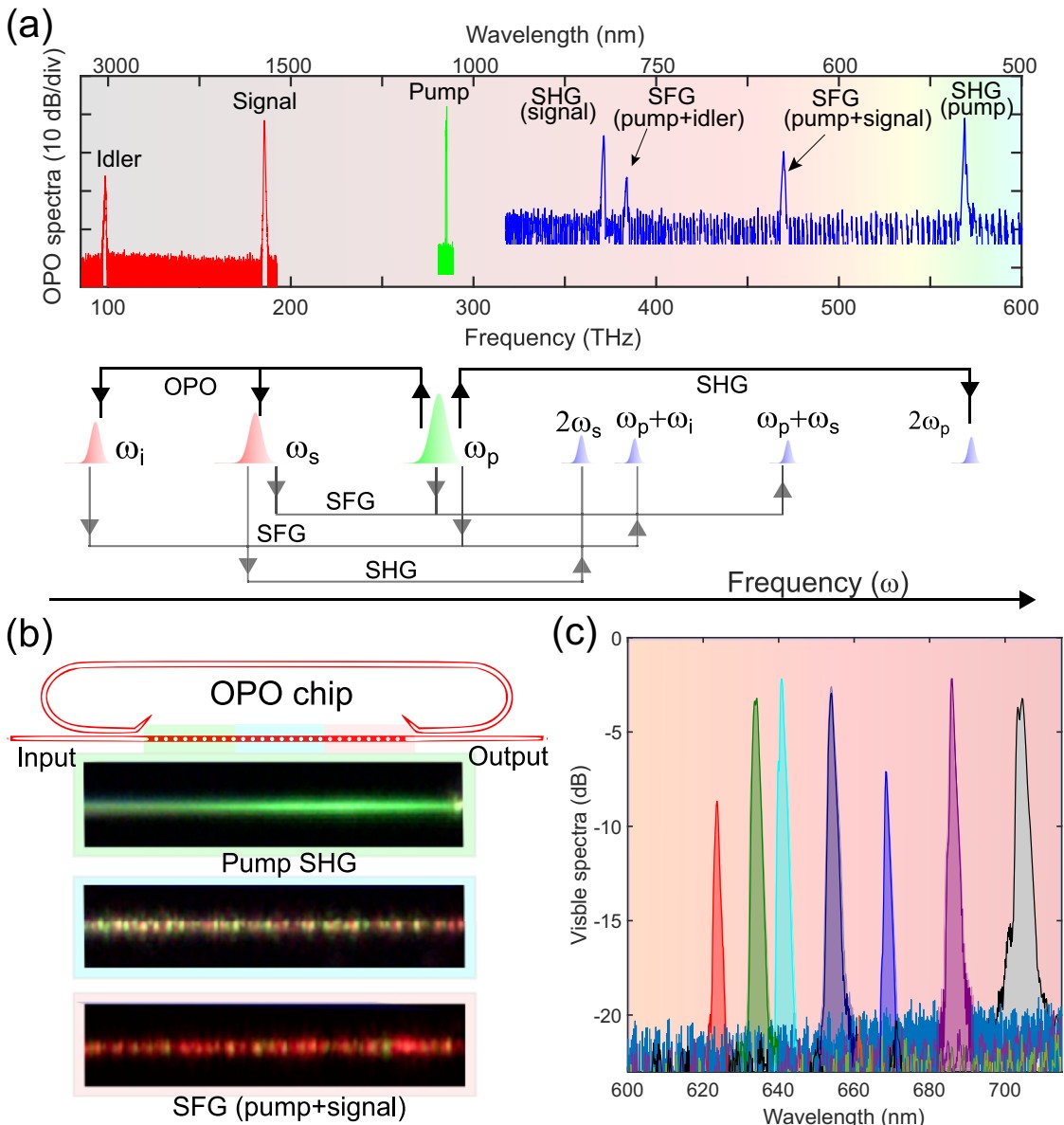

**Fig. 4 | Visible frequency comb generation from the integrated optical parametric oscillator chip. a** The complete emission spectrum of an OPO (Spectra obtained from different optical spectrum analyzers/ spectrometers are stitched together). Apart from the emission of the signal and the idler waves, the OPO also produces output in the visible spectra owing to the auxiliary nonlinear processes, namely the second-harmonic generation (SHG) and the sum-frequency generation (SFG). Schematic representations of the different nonlinear processes are shown. The processes marked by solid black arrows are initiated first, followed by the ones marked in gray. **b** Optical microscope image capturing the visible light emission from various regions of the periodically poled section of the OPO device, **c** Tunable visible frequency comb generation from the integrated OPO chip, where different colors indicate spectra obtained from OPOs with distinct poling periods.

SFG between the pump and the signal waves leads to tunable visible frequency comb generation between 600 and 700 nm, as shown in Fig. 4c. Tuning the OPO farther from degeneracy leads to idler emission further into the mid-IR as well as the SFG component that lies to the bluer side of the visible spectrum.

## Discussion

The pump, which is a near-IR electro-optic comb, can be incorporated into the lithium niobate chip in the future[29,43]. With proper dispersion engineering, our OPO design can additionally achieve a large instantaneous bandwidth accompanied by significant pulse compression[38], enabling the generation of femtosecond mid-infrared frequency combs in nanophotonics. The low power requirement combined with the need for a relatively narrow pump tunability range opens the door

for pumping the OPO chip with butt-coupled near-infrared diode lasers. This paves the way for a fully integrated solution for mid-IR frequency comb generation based on lithium niobate nanophotonics[30,31,44,45] (see Supplementary Section 12). It is worth noting that currently, the conversion efficiencies of the up-conversion processes are substantially lower than the OPO efficiency. The upconversion efficiencies can be enhanced by utilizing appropriate poling periods in the PPLN section. While our current experimental system allows tunability over multiple discrete spectral windows, the demonstrated tuning mechanisms through tuning the phase matching and pump wavelength can yield continuous tuning far beyond the demonstrated range, as discussed in Supplementary Section 15.

In summary, we have demonstrated widely tunable frequency combs covering from the mid-infrared (up to 3.3 μm) to the visible

(620 nm) using nanophotonic OPOs on a single chip. We have shown broadband operation (supporting sub-picosecond pulses) of the OPOs at an electronically congenial repetition rate of ~19 GHz. The signal/idler outputs of the OPOs cover an octave of tuning (from 1.5 to 3.3 μm) enabled by tuning the QPM. The same chip is also capable of generating tunable visible frequency combs owing to the simultaneous occurrence of several ($\chi^{(2)}$) up-conversion processes, giving access to the nearly continuous tuning of frequency combs from visible to mid-IR.

## Methods

### Device fabrication

The devices are fabricated on a 700-nm thick X-cut MgO-doped lithium niobate on silica die (NANOLN). Periodic poling is performed by first patterning electrodes using e-beam lithography, followed by e-beam evaporation of Cr/ Au, and subsequently metal lift-off. Ferroelectric domain inversion is undergone by applying high voltage pulses, and the poling quality is inspected using second-harmonic microscopy. The waveguides are patterned by e-beam lithography and dry-etched with Ar$^+$ plasma. The waveguide facets are polished using fiber polishing films. The OPO-chip consists of multiple devices with poling periods ranging from 5.55 to 5.7 μm (in 10-nm increments) that provide parametric gain spanning over an octave.

### Experimental setup and measurements

The simplified experimental schematic is shown in Fig. 2a, a detailed description of which is provided in Supplementary Sections 4, 5, and 6. Techniques for detecting the carrier envelope offset frequencies and verifying the coherence of the OPO frequency comb are described in Supplementary Sections 10 and 11, respectively. Optical spectra were recorded using a combination of a near-infrared optical spectrum analyzer (OSA) (Yokogawa AQ6374), mid-infrared OSAs (Yokogawa AQ6375B, AQ6376E), and a CCD spectrometer (Thorlabs CCS200). The OPOs are synchronously driven at either the fundamental repetition rate (~9.5 GHz) or its harmonic (~19 GHz). The optical spectrum results are obtained with the harmonic repetition rate operation as it leads to wider instantaneous bandwidth owing to shorter electro-optic pump pulses. The OPOs operating at longer wavelengths have higher thresholds (because of increased effective area, increased coupler loss corresponding to the signal wave, and larger mismatch between the relative walk-off parameters of the signal and the idler wave), and therefore, we operate them intermittently in what we call "quasi-synchronous" operation, as a way to reduce the average power and avoid thermal damage. (see Supplementary Section 5). This limitation is mainly attributed to the avoidable input insertion loss (~12 dB) of our current setup. With the aid of better fiber-to-chip coupling design/mechanisms (insertion loss of the order of 1 dB has been reported in the context of thin-film lithium niobate), the mid-IR OPOs can be operated in a steady state sync-pumped configuration[46]. The phase-matching was verified by observing the second-harmonic generation of an incident 2 μm pulsed source on a waveguide placed next to the OPO and sharing the same poling period.

### System modeling and simulation

We used commercial software (Ansys Lumerical) to solve for the waveguide modes, obtain the tuning/phase-matching curves, as well as to design the adiabatic coupler. For the nonlinear optical simulation, we simulated both the dual envelope model (Supplementary Section 1) and the single envelope model (Supplementary Section 2) using the split-step Fourier algorithm.

## Data availability

Source data are available for this paper and can be found at the Figshare link. All other data that support the plots within this paper and other findings of this study are available from the corresponding author upon reasonable request.

## Code availability

The codes that support the findings of this study are available from the corresponding author upon reasonable request.

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

## Acknowledgements

The device nanofabrication was performed at the Kavli Nanoscience Institute (KNI) at Caltech. This work was supported by a NASA Space Technology Graduate Research Opportunities Award. The authors thank NTT Research for their financial and technical support. The authors thank Dr. Mahmood Bagheri for loaning the Mid-IR optical spectrum analyzer. The authors gratefully acknowledge support from ARO grant no. W911NF-23-1-0048, AFOSR award FA9550-20-1-0040, NSF Grant No. 1846273, and 1918549, NASA, and Center for Sensing to Intelligence at Caltech.

## Author contributions

A.R., L.L., L.C. and R.G. performed the experiments. L.L. fabricated the chip with help from R.S. A.R., L.L. and R.G. performed the numerical simulations. Q.G., M.L. and R.M.B. contributed to the design, discussions, and debugging. A.R. and A.M. wrote the manuscript with input from all authors. A.M. supervised the project.

## Competing interests

L.L., R.M.B., and A.M. are inventors on granted U.S. patent 11,226,538 covering thin-film optical parametric oscillators. L.L., A.M., A.R., R.S., and R.G. are inventors on a U.S. provisional patent application filed by the California Institute of Technology (application number 63/466,188) on 12 May 2023. L.L., A.M., and R.G. are inventors on a U.S. provisional patent application filed by the California Institute of Technology (application number 63/434,015) on 20 December 2022. L.L. and A.M. are involved in developing photonic integrated nonlinear circuits at PINC Technologies Inc. L.L. and A.M. have an equity interest in PINC Technologies Inc. The other authors declare that they have no competing interests.
