## [Peer Review File · Nature Communications]

REVIEWER COMMENTS

Reviewer #1 (Remarks to the Author):

In this paper, the authors presented a scheme for generating a tunable frequency comb in the visible-to-mid IR range using an optical parametric oscillator (OPO) based on thin-film lithium niobate (TFLN). The TFLN device comprises a periodically poled lithium niobate (PPLN) waveguide and a ring resonator. A crucial aspect of this scheme is the utilization of adiabatic couplers within the resonator, which selectively couples long wavelengths to the ring for resonating in the OPO, resulting in doubly resonant conditions for signal and idler generation. The results are relatively in a good match with the explanations. It is essential to address the following inquiries in a clear and concise manner before proceeding further:

1. In the paper, the authors present spectral data using color-filled representations, as shown in Figures 2b, 2c, 3b, 3c, 3d, and 3f. However, it can be challenging to differentiate between the power shape envelope and the comb lines' shape in these figures. To address this issue, it is recommended to employ two different formats to clearly represent these two distinct aspects. This will enhance the understanding of the data presented. Additionally, it would be beneficial to provide information about the linewidth of the OPO spectra and the coherent time, as these details are of interest.

2. To enhance the clarity of the OPO, second harmonic generation (SHG), and sum frequency generation (SFG) processes, it would be beneficial to provide corresponding dynamic illustrations that depict these processes. It would be helpful to include a process sequence diagram in Figure 4a to illustrate the generation relationship among different components.

3. There seems to be a discrepancy in the wavelength between the green optical frequency comb (zoom-in view) and Figure 2c. Please ensure consistency in the wavelength values between these figures.

4. In addition to the OPO, SHG and SFG also play significant roles in wavelength conversion. It would be beneficial if the authors could provide more detailed information about these two processes, including their conversion efficiency. Furthermore, when calculating the conversion efficiency of the OPO, it is important to clarify whether the processes of SHG and SFG are included in the calculations. Additionally, it would be helpful to explain how one can determine whether SHG and SFG processes are considered in the conversion efficiency assessment.

5. How to control the switch from non-degenerate OPO to different degenerate OPO states?

6. What are the advantages of using a fluoride fiber at the output of the OPO chip? Please elaborate on this aspect.

7. Temperature variations can also affect the phase matching curve. It would be valuable if the authors could provide the corresponding measured results and compare them with the results obtained by varying the poling period. This comparison would help to understand the relative impact of temperature and poling period changes on the phase matching characteristics.

Addressing these questions and concerns will improve the logical structure, clarity, and overall coherence of the paper

Reviewer #2 (Remarks to the Author):

In this paper, the authors demonstrated the generation of frequency combs deep into the mid-IR spectrum through a well-designed OPO process that converts an input ps telecom EO comb into longer wavelength signal and idler pulses. This process is made both efficient and widely tunable thanks to the use of a PPLN-embedded resonator that is designed to only resonate at the signal and idler frequencies, but allows the pump wavelength to pass through only once, via a carefully engineered adiabatic coupler. As a result, the authors were able to demonstrate the generation of combs at wavelengths as long as 3.3 μm , which is a spectral region typically highly nontrivial to achieve. Additionally, the authors utilized SFG/SHG process between pump/signal/idler to further demonstrate the generation of frequency combs in the visible spectrum. Overall, I find the doubly-resonant cavity design interesting, the spectral coverage of the generated combs impressive, and the requirement of only a ps EO comb source practically attractive, and I believe the results would be of interest to the broad audience in Nature Communications. I would therefore recommend the paper to be published if the following concern/comments can be properly addressed.

1. My main concern after reading the paper is the claimed "tunability" or "tuning range" of the device. From the title, abstract, and perhaps also Fig. 1, the impression I had was that the device is able to generate combs that covers spectral ranges all the way from telecom to mid-IR continuously. For the sake of simplicity, we can leave out the discussions of SHG/SFG and focus on the OPO process for now. At least from Fig. 1b (bottom panel), theoretically the devices should be able to cover OPO output continuously from 1500 nm to 3300 nm. However, if I understand correctly, the actually measured devices only cover certain discrete regions (e.g. 1890-1970 nm in Fig. 3c) within this full range. If my understanding is incorrect, I'd like to see a figure that shows the combs generated all over the full spectrum (e.g. using a waferfall figure). Otherwise, I hope the authors could discuss in more details what main reasons have prevented the OPO process to take place in certain wavelength ranges, and also to

tone down some of the descriptions in the text, e.g. in abstract ("tunable" beyond an octave); in summary ("cover an octave of tuning"); and possibly in other places. In the end of the day, if a discrete coverage can be called "tunable over an octave", then in principle I can achieve this by only generating two combs at the starting/ending wavelengths.

2. Similarly, I also have some doubts of calling this "tunable" by switching between different devices. Perhaps this can be rephrased as something like "achieving XX spectral coverage by devices fabricated on the same photonic chip". Overall I believe the results in this work is already quite impressive and I suggest to make these facts clearer and less confusing.

Some other minor comments/questions:

3. Second paragraph, line 6, I'd probably not say "comb power distributed over a wide frequency range" is a bad thing, although it is true that different applications would prefer combs with different spans. The fact that the combs in this work has a smaller spectral bandwidth is more of a characteristic rather than an advantage.

4. P3, paragraph 3, line 5-8, as I understand the octave-wide tunability (and here please be careful too) is achieved by both pump wavelength tuning and switching between different devices, right?

5. P3, right column, paragraph 2, line 13-15, I'm not too sure why the green light would be overpowered by red when passing through the poled region, as both components are generated via the PPLN I suppose. Please clarify.

Response to Reviewers

Reviewer 1

In this paper, the authors presented a scheme for generating a tunable frequency comb in the visible-to-mid IR range using an optical parametric oscillator (OPO) based on thin-film lithium niobate (TFLN). The TFLN device comprises a periodically poled lithium niobate (PPLN) waveguide and a ring resonator. A crucial aspect of this scheme is the utilization of adiabatic couplers within the resonator, which selectively couples long wavelengths to the ring for resonating in the OPO, resulting in doubly resonant conditions for signal and idler generation. The results are relatively in a good match with the explanations. It is essential to address the following inquiries in a clear and concise manner before proceeding further:

We would like to thank the reviewer for the careful review of our manuscript. We appreciate that the reviewer highlighted the use of adiabatic coupler which forms an integral part of our demonstration that enabled wide range tunability from visible to mid-IR. Apart from this wide tunability, we would like to emphasize that our work also represents the first demonstration of synchronously pumped OPO in an integrated $\chi^{(2)}$ nonlinear platform. We have revised our manuscript taking into consideration the comments and suggestions provided by the reviewer. Below we provide a point-by-point response to the reviewer's concerns.

In the paper, the authors present spectral data using color-filled representations, as shown in Figures 2b, 2c, 3b, 3c, 3d, and 3f. However, it can be challenging to differentiate between the power shape envelope and the comb lines' shape in these figures. To address this issue, it is recommended to employ two different formats to clearly represent these two distinct aspects. This will enhance the understanding of the data presented. Additionally, it would be beneficial to provide information about the linewidth of the OPO spectra and the coherent time, as these details are of interest.

Most of the spectral data presented in the main section of the manuscript is represented in terms of the power spectrum envelope. The only exception applies to the insets of Fig. 2(c), where the comb lines of the spectrum are resolved (but still limited by the optical spectrum analyzer resolution). We understand that the two representations look similar and may confuse readers. This is especially because we have color-filled the area under the power spectrum envelope to clearly distinguish the spectrum originating from distinct OPOs. To avoid potential confusion, we have added this line to the caption of Figure 2:

Zoomed-in versions display the underlying comb line structure of the power spectrum envelope of the signal and idler combs.

We thank the reviewer for asking to provide more information about the linewidth of the OPO spectrum. To further investigate the coherence, we performed field cross-correlation measurements which are shown in Fig. 3(e,f). Further details regarding the linewidth of the OPO signal and idler combs are provided in supplementary sections 10 and 11.

To enhance the clarity of the OPO, second harmonic generation (SHG), and sum frequency generation (SFG) processes, it would be beneficial to provide corresponding dynamic illustrations that depict these processes. It would be helpful to include a process sequence diagram in Figure 4a to illustrate the generation relationship among different components.

We thank the reviewer for this suggestion. We have now incorporated a process sequence diagram (Fig. 4(a) lower panel) to illustrate the underlying processes.

There seems to be a discrepancy in the wavelength between the green optical frequency comb (zoom-in view) and Figure 2c. Please ensure consistency in the wavelength values between these figures.

We thank the reviewer for bringing this to our attention. We inadvertently made this mistake where we used the wrong coloring scheme to highlight the comb line spectrum. This is rectified now.

In addition to the OPO, SHG and SFG also play significant roles in wavelength conversion. It would be beneficial if the authors could provide more detailed information about these two processes, including their conversion efficiency. Furthermore, when calculating the conversion efficiency of the OPO, it is important to clarify whether the processes of SHG and SFG are included in the calculations. Additionally, it would be helpful to explain how one can determine whether SHG and SFG processes are considered in the conversion efficiency assessment.

We thank the reviewer for this comment. The simplified fit to the OPO conversion efficiency (see Figure 6(a) of supplementary section 3) doesn't incorporate the SHG and SFG mechanisms. We have added this clarification in the supplementary section 3 where we present the efficiencies. The reviewer is absolutely right, that the SFG and SHG processes will in fact impact the OPO signal and idler conversion efficiencies. However, we note that the current poled region only provides phase matching for the OPO process and hence the conversion efficiencies of the other processes are substantially lower than the OPO efficiency. This is consistent with the numerical simulations in SI section 2. To emphasize this aspect we have added this line to the main text:

It is worth noting that currently, the conversion efficiencies of these up-conversion processes are substantially lower than the OPO efficiency. The upconversion efficiencies can be enhanced by utilizing appropriate poling periods in the PPLN section.

How to control the switch from non-degenerate OPO to different degenerate OPO states?

We thank the reviewer for this important question. The crucial parameters in a doubly-resonant OPO to determining operation in the degenerate or non-degenerate regimes are the parametric gain spectrum and the cavity detuning as detailed in our earlier work in a different experimental platform [1].

The parameter tuning knobs we have access to are poling period variation (which is lithographically defined and fixed post-fabrication), repetition-rate tuning (which can be varied by changing the repetition rate of the EO-comb around the cavity FSR), pump wavelength tuning, and temperature tuning. Each of these parameters individually or a combination of them can affect the cavity detuning and round-trip delay mismatch that is responsible for setting the OPO operation point.

What are the advantages of using a fluoride fiber at the output of the OPO chip? Please elaborate on this aspect.

We thank the reviewer for raising this question. We resorted to Fluoride fiber since it has broadband transparency in the mid-infrared. The usual silica-based fiber (those are not specially processed) suffers from the absorption band of silica centered around 2.7 μm . For example, one of the vendors of optical fibers (Thorlabs), quotes the specifications of their standard silica-based fibers to be usable till 2.2 μm without incurring excessive loss.

Temperature variations can also affect the phase matching curve. It would be valuable if the authors could provide the corresponding measured results and compare them with the results obtained by varying the poling period. This comparison would help to understand the relative impact of temperature and poling period changes on the phase matching characteristics.

We thank the reviewer for raising this issue. We have analyzed the effect of temperature variations on the phase-matching in supplementary section 13. We have performed some experimental measurements with temperature variation, and indeed observed the shift in the OPO output. To further elaborate on temperature tuning, we have added new results in the supplementary (section 13). We present the effect of temperature variation on the phase-matching in periodically poled thin-film nanophotonic devices. Figure 19 (supplementary (section 13)) shows how the optical parametric generation spectrum is affected by temperature variation. Results obtained by numerical simulation agree with the experimentally obtained data [2], confirming the accuracy of our temperature tuning model.

While we are confident about the temperature tuning of phase matching, systematic measurement of OPO output as a function of temperature is challenging with our current experimental setup. This is mainly because when we resort to changing the temperature of the chip, not only is the phase-matching in the periodically poled waveguide altered, but the ring cavity also experiences modification in its group index which in turn alters the FSR of the cavity. It is a cumbersome process to modify the EO comb pump for every setting of the temperature change which is required for a systematic study. Our EO comb comprises of a cascade of three phase modulators which are driven by amplified RF signal oscillating at the FSR frequency. This set of 3 amplifiers are preceded by corresponding phase delay units that are required to ensure all the phase modulators are driven in-phase. Every different setting of FSR enforces a different setting for these phase delay units and also a unique setting for the waveshaper. This is the reason we avoided the systematic study of temperature variations.

This can be mitigated in our next realizations of OPOs, where we can utilize local heating components in the cavity with independent FSR tuning mechanisms. Moreover, we expect that the utilization of other ps pump sources, such as our recently demonstrated mode-locked laser [3] can eliminate the potential challenges associated with an EO comb pump consisting of a cascade of modulators.

Addressing these questions and concerns will improve the logical structure, clarity, and overall coherence of the paper.

We thank the reviewer for the careful review of the manuscript and the constructive feedback. Based on the reviewer's comments and suggestions, we have revised the manuscript addressing all the concerns raised.

Reviewer 2

In this paper, the authors demonstrated the generation of frequency combs deep into the mid-IR spectrum through a well-designed OPO process that converts an input ps telecom EO comb into longer wavelength signal and idler pulses. This process is made both efficient and widely tunable thanks to the use of a PPLN-embedded resonator that is designed to only resonate at the signal and idler frequencies, but allows the pump wavelength to pass through only once, via a carefully engineered adiabatic coupler. As a result, the authors were able to demonstrate the generation of combs at wavelengths as long as 3.3 μm , which is a spectral region typically highly nontrivial to achieve. Additionally, the authors utilized SFG/SHG process between pump/signal/idler to further demonstrate the generation of frequency combs in the visible spectrum. Overall, I find the doubly-resonant cavity design interesting, the spectral coverage of the generated combs impressive, and the requirement of only a ps EO comb source practically attractive, and I believe the results would be of interest to the broad audience in Nature Communications. I would therefore recommend the paper to be published if the following concern/comments can be properly addressed.

We would like to thank the reviewer for the careful review of our revised manuscript. We thank the reviewer for highlighting the major contributions of our present work and the positive comments. We have revised our manuscript taking into consideration the comments and suggestions provided by the reviewer. Below we provide answers to all the reviewer's concerns.

My main concern after reading the paper is the claimed "tunability" or "tuning range" of the device. From the title, abstract, and perhaps also Fig. 1, the impression I had was that the device is able to generate combs that covers spectral ranges all the way from telecom to mid-IR continuously. For the sake of simplicity, we can leave out the discussions of SHG/SFG and focus on the OPO process for now. At least from Fig. 1b (bottom panel), theoretically the devices should be able to cover OPO output continuously from 1500 nm to 3300 nm. However, if I understand correctly, the actually measured devices only cover certain discrete regions (e.g. 1890-1970 nm in Fig. 3c) within this full range. If my understanding is incorrect, I'd like to see a figure that shows the combs generated all over the full spectrum (e.g. using a waterfall figure). Otherwise, I hope the authors could discuss in more details what main reasons have prevented the OPO process to take place in certain wavelength ranges, and also to tone down some of the descriptions in the text, e.g. in abstract ("tunable" beyond an octave); in summary ("cover an octave of tuning"); and possibly in other places. In the end of the day, if a discrete coverage can be called "tunable over an octave", then in principle I can achieve this by only generating two combs at the starting/ending wavelengths.

We thank the reviewer for raising this important concern. First, we would like to emphasize that, as the reviewer noted, the presented device allows for continuous tuning of the OPO output comb beyond an octave. However, experimental demonstration of such a continuous tuning over this unprecedented tuning range is challenging in our current experimental setup. In the current manuscript, we show a large number of spectra (far beyond Fig. 3c) to unequivocally demonstrate such a capability through two main tuning mechanisms.

The first mechanism is through tuning the phase matching. This is currently achieved experimentally by switching the poling periods. This has led to coarse tunings of the OPO outputs as shown in Fig. 2. We further establish in supplementary section 13 through numerical simulations that even with one poling period, one can achieve the whole octave tuning with temperature tuning of the poled region. However, such a temperature tuning is challenging with our current experimental setup as we explain below.

The second tuning mechanism provided by the OPO is through tuning the pump central wavelengths. For each poling period, we have shown experimentally that this mechanism can yield fine-tuning of the output spectrum. An example of such tuning is shown in Fig. 3c. More tuning data obtained experimentally are provided in supplementary section 14 (for infrared) and supplementary section 2 (for visible and near-infrared).

With these two tuning mechanisms, our experimental results establish the capability of the device to provide continuous tuning beyond an octave. Here, we would like to point out the experimental challenges associated with generating a single waterfall figure to cover the whole tuning spectrum.

- **Current experimental limitations in pump tuning:** This limitation pertains to our current utilization of a YDFA amplifier to boost the power levels of the pump (EO comb) and overcome the insertion loss. The YDFA has a gain window which constrained us to operate in that spectral window. Moreover, the waveshaper we used also imposed an upper cut-off wavelength of operation. Both of these combined prevented us from leveraging a wider range of pump wavelength tuning which would give us a much broader tuning range from a single OPO.
- **Current limitation on temperature tuning:** While we are confident about the temperature tuning of phase matching, systematic measurement of OPO output as a function of temperature is challenging with our current experimental setup. This is mainly because when we resort to changing the temperature of the chip, not only is the phase-matching in the periodically poled waveguide altered, but the ring cavity also experiences modification in its group index which in turn alters the FSR of the cavity. It is a cumbersome process to modify the EO comb pump for every setting of the temperature change which is required for a systematic study. Our EO comb comprises of a cascade of three phase modulators which are driven by amplified RF signal oscillating at the FSR frequency. This set of 3 amplifiers are preceded by corresponding phase delay units that are required to ensure all the phase modulators are driven in-phase. Every different setting of FSR enforces a different

setting for these phase delay units and also a unique setting for the waveshaper. This is the reason we avoided the systematic study of temperature variations.

- Our lithium niobate on insulator chip uses a silica layer as an insulator. The silica is known to have an absorption band around $2.7 \mu\text{m}$. This absorption rendered OPO threshold to be very high to see oscillation in this spectral band. Thus we won't expect to see continuous OPO output coverage in this region even though the two previous limitations are eliminated. The solution to this issue is to use a lithium niobate on a sapphire wafer. The sapphire has broadband transparency and is more suitable for mid-IR applications. Our recent devices use LNOI with sapphire as the insulating layer.

These limitations can be mitigated in our next OPO designs by using local heating components in the cavity with independent FSR tuning mechanisms. Moreover, we expect that utilization of other ps pump sources, such as our recently demonstrated mode-locked laser [3] can eliminate the limitations associated with an EO comb pump.

To avoid potential confusion and misinterpretation of our results, we have added this sentence to the discussion section: *While our current experimental system allows tunability over multiple discrete spectral windows the demonstrated tuning mechanisms through tuning the phase matching and pump wavelength can yield continuous tuning far beyond the demonstrated range as discussed in supplementary section 15.*

Similarly, I also have some doubts of calling this "tunable" by switching between different devices. Perhaps this can be rephrased as something like "achieving XX spectral coverage by devices fabricated on the same photonic chip". Overall I believe the results in this work is already quite impressive and I suggest to make these facts clearer and less confusing.

We thank the reviewer for the suggestion. We have added more information to Fig. 2(c) to clarify this aspect. By switching to different devices, we refer to the coarse tuning (i.e. changing to a different OPO fabricated on the same chip with its distinct poling period) which is now distinguished by numbering the OPOs (OPO 1 to 7).

The fine-tuning (achieved via pump wavelength tuning) is now shown with a combination of filled dots/ open circles/ dashed lines with the respective spectrum data presented in the supplementary information.

Some other minor comments/questions:

Second paragraph, line 6, I'd probably not say "comb power distributed over a wide frequency range" is a bad thing, although it is true that different applications would prefer combs with different spans. The fact that the combs in this work has a smaller spectral bandwidth is more of a characteristic rather than an advantage.

We thank the reviewer for the suggestion. We have added a new sentence to clarify this. The new sentence reads as:

While the broad instantaneous spectral bandwidth maybe suitable for certain applications, others may require the existence of more concentrated spectral power to enhance the signal-to-noise ratio.

At the same time, we would like to mention, that our OPO can also possess wide instantaneous bandwidth. This would require careful dispersion engineering namely the group velocity dispersion and group velocity mismatch parameters. To simultaneously achieve tunability and broad instantaneous bandwidth however changing the poling period is not sufficient. In that case, one has to tailor the dispersion of the individual OPOs to modify the dispersion in the spectral band that it is catering to.

P3, paragraph 3, line 5-8, as I understand the octave-wide tunability (and here please be careful too) is achieved by both pump wavelength tuning and switching between different devices, right?

Yes. The octave-wide tunability is achieved by a combination of pump wavelength tuning and switching between different devices (on the same chip), i.e. changing the poling period. This is also evident from our illustration of Fig. 1(b) where the wide tunability is expected from a combination of both the pump wavelength tuning and switching the OPO device.

P3, right column, paragraph 2, line 13-15, I'm not too sure why the green light would be overpowered by red when passing through the poled region, as both components are generated via the PPLN I suppose. Please clarify.

We would like to thank the reviewer for raising this question. The efficiency of the generated visible components is a function of the parasitic phase-matching originating due to random poling duty-cycle variations. The comment above pertains to the visual perception as observed from the scattered light emanating from the waveguide. The scattering is associated with the unavoidable surface roughness of the waveguide walls. Thus the scattered intensity is a complex interplay of surface roughness and conversion efficiency. We agree that this particular observation maybe specific to the OPO under consideration and may not be a universal feature.

References

- [1] Arkadev Roy, Saman Jahani, Carsten Langrock, Martin Fejer, and Alireza Marandi. Spectral phase transitions in optical parametric oscillators. *Nature communications*, 12(1):835, 2021.
- [2] Luis Ledezma, Ryoto Sekine, Qiushi Guo, Rajveer Nehra, Saman Jahani, and Alireza Marandi. Intense optical parametric amplification in dispersion-engineered nanophotonic lithium niobate waveguides. *Optica*, 9(3):303–308, 2022.
- [3] Qiushi Guo, Ryoto Sekine, James A Williams, Benjamin K Gutierrez, Robert M Gray, Luis Ledezma, Luis Costa, Arkadev Roy, Selina Zhou, Mingchen Liu, et al. Mode-locked laser in nanophotonic lithium niobate. *arXiv preprint arXiv:2306.05314*, 2023.

REVIEWERS' COMMENTS

Reviewer #1 (Remarks to the Author):

The authors have answered and solved my questions and doubts very well, and I recommend accepting this paper.

Reviewer #2 (Remarks to the Author):

The authors have addressed all my comments.

Response to Reviewers

Reviewer 1

The authors have answered and solved my questions and doubts very well, and I recommend accepting this paper.

We would like to thank the reviewer for the careful review of our manuscript. We are glad to be able to answer all the reviewer's comments and questions.

Reviewer 2

The authors have addressed all my comments.

We would like to thank the reviewer for the careful review of our manuscript. We are glad to be able to answer all the reviewer's comments and questions.